# Understanding Deep Neural Networks with Rectified Linear Units

**Raman Arora**[*] **Amitabh Basu**[†] **Poorya Mianjy**[‡] **Anirbit Mukherjee**[§]
Johns Hopkins University

## Abstract

In this paper we investigate the family of functions representable by deep neural networks (DNN) with rectified linear units (ReLU). We give an algorithm to train a ReLU DNN with one hidden layer to *global optimality* with runtime polynomial in the data size albeit exponential in the input dimension. Further, we improve on the known lower bounds on size (from exponential to super exponential) for approximating a ReLU deep net function by a shallower ReLU net. Our gap theorems hold for smoothly parametrized families of "hard" functions, contrary to countable, discrete families known in the literature. An example consequence of our gap theorems is the following: for every natural number $k$ there exists a function representable by a ReLU DNN with $k^2$ hidden layers and total size $k^3$, such that any ReLU DNN with at most $k$ hidden layers will require at least $\frac{1}{2}k^{k+1} - 1$ total nodes. Finally, for the family of $\mathbb{R}^n \to \mathbb{R}$ DNNs with ReLU activations, we show a new lowerbound on the number of affine pieces, which is larger than previous constructions in certain regimes of the network architecture and most distinctively our lowerbound is demonstrated by an explicit construction of a *smoothly parameterized* family of functions attaining this scaling. Our construction utilizes the theory of zonotopes from polyhedral theory.

## 1 Introduction

Deep neural networks (DNNs) provide an excellent family of hypotheses for machine learning tasks such as classification. Neural networks with a single hidden layer of finite size can represent any continuous function on a compact subset of $\mathbb{R}^n$ arbitrary well. The universal approximation result was first given by Cybenko in 1989 for sigmoidal activation function (Cybenko, 1989), and later generalized by Hornik to an arbitrary bounded and nonconstant activation function Hornik (1991). Furthermore, neural networks have finite VC dimension (depending polynomially on the number of edges in the network), and therefore, are PAC (probably approximately correct) learnable using a sample of size that is polynomial in the size of the networks Anthony & Bartlett (1999). However, neural networks based methods were shown to be computationally hard to learn (Anthony & Bartlett, 1999) and had mixed empirical success. Consequently, DNNs fell out of favor by late 90s.

Recently, there has been a resurgence of DNNs with the advent of deep learning LeCun et al. (2015). Deep learning, loosely speaking, refers to a suite of computational techniques that have been developed recently for training DNNs. It started with the work of Hinton et al. (2006), which gave empirical evidence that if DNNs are initialized properly (for instance, using unsupervised pre-training), then we can find good solutions in a reasonable amount of runtime. This work was soon followed by a series of early successes of deep learning at significantly improving the state-of-the-art in speech recognition Hinton et al. (2012). Since then, deep learning has received immense attention from the machine learning community with several state-of-the-art AI systems in speech recognition, image classification, and natural language processing based on deep neural nets Hinton et al. (2012); Dahl et al. (2013); Krizhevsky et al. (2012); Le (2013); Sutskever et al. (2014). While there is less of evidence now that pre-training actually helps, several other solutions have since been put forth

[*]Department of Computer Science, Email: `arora@cs.jhu.edu`

[†]Department of Applied Mathematics and Statistics, Email: `basu.amitabh@jhu.edu`

[‡]Department of Computer Science, Email: `mianjy@jhu.edu`

[§]Department of Applied Mathematics and Statistics, Email: `amukhe14@jhu.edu`

to address the issue of efficiently training DNNs. These include heuristics such as dropouts Srivastava et al. (2014), but also considering alternate deep architectures such as convolutional neural networks Sermanet et al. (2014), deep belief networks Hinton et al. (2006), and deep Boltzmann machines Salakhutdinov & Hinton (2009). In addition, deep architectures based on new non-saturating activation functions have been suggested to be more effectively trainable – the most successful and widely popular of these is the rectified linear unit (ReLU) activation, i.e., $\sigma(x) = \max\{0, x\}$, which is the focus of study in this paper.

In this paper, we formally study deep neural networks with rectified linear units; we refer to these deep architectures as ReLU DNNs. Our work is inspired by these recent attempts to understand the reason behind the successes of deep learning, both in terms of the structure of the functions represented by DNNs, Telgarsky (2015; 2016); Kane & Williams (2015); Shamir (2016), as well as efforts which have tried to understand the non-convex nature of the training problem of DNNs better Kawaguchi (2016); Haeffele & Vidal (2015). Our investigation of the function space represented by ReLU DNNs also takes inspiration from the classical theory of circuit complexity; we refer the reader to Arora & Barak (2009); Shpilka & Yehudayoff (2010); Jukna (2012); Saptharishi (2014); Allender (1998) for various surveys of this deep and fascinating field. In particular, our gap results are inspired by results like the ones by Hastad Hastad (1986), Razborov Razborov (1987) and Smolensky Smolensky (1987) which show a strict separation of complexity classes. We make progress towards similar statements with deep neural nets with ReLU activation.

## 1.1 NOTATION AND DEFINITIONS

We extend the ReLU activation function to vectors $x \in \mathbb{R}^n$ through entry-wise operation: $\sigma(x) = (\max\{0, x_1\}, \max\{0, x_2\}, \ldots, \max\{0, x_n\})$. For any $(m, n) \in \mathbb{N}$, let $\mathcal{A}_m^n$ and $\mathcal{L}_m^n$ denote the class of affine and linear transformations from $\mathbb{R}^m \to \mathbb{R}^n$, respectively.

**Definition 1.** [ReLU DNNs, depth, width, size] For any *number of hidden layers* $k \in \mathbb{N}$, *input and output dimensions* $w_0, w_{k+1} \in \mathbb{N}$, a $\mathbb{R}^{w_0} \to \mathbb{R}^{w_{k+1}}$ ReLU DNN is given by specifying a sequence of $k$ natural numbers $w_1, w_2, \ldots, w_k$ representing *widths* of the hidden layers, a set of $k$ affine transformations $T_i : \mathbb{R}^{w_{i-1}} \to \mathbb{R}^{w_i}$ for $i = 1, \ldots, k$ and a linear transformation $T_{k+1} : \mathbb{R}^{w_k} \to \mathbb{R}^{w_{k+1}}$ corresponding to *weights* of the hidden layers. Such a ReLU DNN is called a $(k + 1)$-layer ReLU DNN, and is said to have $k$ hidden layers. The function $f : \mathbb{R}^{n_1} \to \mathbb{R}^{n_2}$ computed or represented by this ReLU DNN is

$$f = T_{k+1} \circ \sigma \circ T_k \circ \cdots \circ T_2 \circ \sigma \circ T_1, \tag{1.1}$$

where $\circ$ denotes function composition. The *depth* of a ReLU DNN is defined as $k + 1$. The *width* of a ReLU DNN is $\max\{w_1, \ldots, w_k\}$. The *size* of the ReLU DNN is $w_1 + w_2 + \ldots + w_k$.

**Definition 2.** We denote the class of $\mathbb{R}^{w_0} \to \mathbb{R}^{w_{k+1}}$ ReLU DNNs with $k$ hidden layers of widths $\{w_i\}_{i=1}^k$ by $\mathcal{F}_{\{w_i\}_{i=0}^{k+1}}$, i.e.

$$\mathcal{F}_{\{w_i\}_{i=0}^{k+1}} := \{T_{k+1} \circ \sigma \circ T_k \circ \cdots \circ \sigma \circ T_1 : T_i \in \mathcal{A}_{w_{i-1}}^{w_i} \forall i \in \{1, \ldots, k\}, \quad T_{k+1} \in \mathcal{L}_{w_k}^{w_{k+1}}\} \tag{1.2}$$

**Definition 3.** [Piecewise linear functions] We say a function $f : \mathbb{R}^n \to \mathbb{R}$ is *continuous piecewise linear (PWL)* if there exists a *finite* set of polyhedra whose union is $\mathbb{R}^n$, and $f$ is affine linear over each polyhedron (note that the definition automatically implies continuity of the function because the affine regions are closed and cover $\mathbb{R}^n$, and affine functions are continuous). The *number of pieces of $f$* is the number of maximal connected subsets of $\mathbb{R}^n$ over which $f$ is affine linear (which is finite).

Many of our important statements will be phrased in terms of the following simplex.

**Definition 4.** Let $M > 0$ be any positive real number and $p \geq 1$ be any natural number. Define the following set:

$$\Delta_M^p := \{\mathbf{x} \in \mathbb{R}^p : 0 < \mathbf{x}_1 < \mathbf{x}_2 < \ldots < \mathbf{x}_p < M\}.$$

## 2 EXACT CHARACTERIZATION OF FUNCTION CLASS REPRESENTED BY RELU DNNS

One of the main advantages of DNNs is that they can represent a large family of functions with a relatively small number of parameters. In this section, we give an exact characterization of the

functions representable by ReLU DNNs. Moreover, we show how structural properties of ReLU DNNs, specifically their depth and width, affects their expressive power. It is clear from definition that any function from $\mathbb{R}^n \to \mathbb{R}$ represented by a ReLU DNN is a continuous piecewise linear (PWL) function. In what follows, we show that the converse is also true, that is any PWL function is representable by a ReLU DNN. In particular, the following theorem establishes a one-to-one correspondence between the class of ReLU DNNs and PWL functions.

**Theorem 2.1.** Every $\mathbb{R}^n \to \mathbb{R}$ ReLU DNN represents a piecewise linear function, and every piecewise linear function $\mathbb{R}^n \to \mathbb{R}$ can be represented by a ReLU DNN with at most $\lceil \log_2(n+1) \rceil + 1$ depth.

**Proof Sketch:** It is clear that any function represented by a ReLU DNN is a PWL function. To see the converse, we first note that any PWL function can be represented as a linear combination of piecewise linear convex functions. More formally, by Theorem 1 in (Wang & Sun, 2005), for every piecewise linear function $f : \mathbb{R}^n \to \mathbb{R}$, there exists a finite set of affine linear functions $\ell_1, \ldots, \ell_k$ and subsets $S_1, \ldots, S_p \subseteq \{1, \ldots, k\}$ (not necessarily disjoint) where each $S_i$ is of cardinality at most $n+1$, such that

$$f = \sum_{j=1}^{p} s_j \left( \max_{i \in S_j} \ell_i \right), \tag{2.1}$$

where $s_j \in \{-1, +1\}$ for all $j = 1, \ldots, p$. Since a function of the form $\max_{i \in S_j} \ell_i$ is a piecewise linear convex function with at most $n+1$ pieces (because $|S_j| \leq n+1$), Equation (2.1) says that any continuous piecewise linear function (not necessarily convex) can be obtained as a linear combination of piecewise linear convex functions each of which has at most $n+1$ affine pieces. Furthermore, Lemmas D.1, D.2 and D.3 in the Appendix (see supplementary material), show that composition, addition, and pointwise maximum of PWL functions are also representable by ReLU DNNs. In particular, in Lemma D.3 we note that $\max\{x, y\} = \frac{x+y}{2} + \frac{|x-y|}{2}$ is implementable by a two layer ReLU network and use this construction in an inductive manner to show that maximum of $n+1$ numbers can be computed using a ReLU DNN with depth at most $\lceil \log_2(n+1) \rceil$.

While Theorem 2.1 gives an upper bound on the depth of the networks needed to represent all continuous piecewise linear functions on $\mathbb{R}^n$, it does not give any tight bounds on the *size* of the networks that are needed to represent a given piecewise linear function. For $n = 1$, we give tight bounds on size as follows:

**Theorem 2.2.** Given any piecewise linear function $\mathbb{R} \to \mathbb{R}$ with $p$ pieces there exists a 2-layer DNN with at most $p$ nodes that can represent $f$. Moreover, any 2-layer DNN that represents $f$ has size at least $p - 1$.

Finally, the main result of this section follows from Theorem 2.1, and well-known facts that the piecewise linear functions are dense in the family of compactly supported continuous functions and the family of compactly supported continuous functions are dense in $L^q(\mathbb{R}^n)$ (Royden & Fitzpatrick, 2010)). Recall that $L^q(\mathbb{R}^n)$ is the space of Lebesgue integrable functions $f$ such that $\int |f|^q d\mu < \infty$, where $\mu$ is the Lebesgue measure on $\mathbb{R}^n$ (see Royden Royden & Fitzpatrick (2010)).

**Theorem 2.3.** Every function in $L^q(\mathbb{R}^n)$, $(1 \leq q \leq \infty)$ can be arbitrarily well-approximated in the $L^q$ norm (which for a function $f$ is given by $||f||_q = (\int |f|^q)^{1/q}$) by a ReLU DNN function with at most $\lceil \log_2(n+1) \rceil$ hidden layers. Moreover, for $n = 1$, any such $L^q$ function can be arbitrarily well-approximated by a 2-layer DNN, with tight bounds on the size of such a DNN in terms of the approximation.

Proofs of Theorems 2.2 and 2.3 are provided in Appendix A. We would like to remark that a weaker version of Theorem 2.1 was observed in (Goodfellow et al., 2013, Proposition 4.1) (with no bound on the depth), along with a universal approximation theorem (Goodfellow et al., 2013, Theorem 4.3) similar to Theorem 2.3. The authors of Goodfellow et al. (2013) also used a previous result of Wang (Wang, 2004) for obtaining their result. In a subsequent work Boris Hanin (Hanin, 2017) has, among other things, found a width and depth upper bound for ReLU net representation of positive PWL functions on $[0,1]^n$. The width upperbound is n+3 for general positive PWL functions and $n+1$ for convex positive PWL functions. For convex positive PWL functions his depth upper bound is sharp if we disallow dead ReLUs.

## 3 BENEFITS OF DEPTH

Success of deep learning has been largely attributed to the depth of the networks, i.e. number of successive affine transformations followed by nonlinearities, which is shown to be extracting hierarchical features from the data. In contrast, traditional machine learning frameworks including support vector machines, generalized linear models, and kernel machines can be seen as instances of shallow networks, where a linear transformation acts on a single layer of nonlinear feature extraction. In this section, we explore the importance of depth in ReLU DNNs. In particular, in Section 3.1, we provide a smoothly parametrized family of $\mathbb{R} \to \mathbb{R}$ "hard" functions representable by ReLU DNNs, which requires exponentially larger size for a shallower network. Furthermore, in Section 3.2, we construct a continuum of $\mathbb{R}^n \to \mathbb{R}$ "hard" functions representable by ReLU DNNs, which to the best of our knowledge is the first explicit construction of ReLU DNN functions whose number of affine pieces grows exponentially with input dimension. The proofs of the theorems in this section are provided in Appendix B.

### 3.1 CIRCUIT LOWER BOUNDS FOR $\mathbb{R} \to \mathbb{R}$ ReLU DNNs

In this section, we are only concerned about $\mathbb{R} \to \mathbb{R}$ ReLU DNNs, i.e. both input and output dimensions are equal to one. The following theorem shows the depth-size trade-off in this setting.

**Theorem 3.1.** For every pair of natural numbers $k \geq 1$, $w \geq 2$, there exists a family of hard functions representable by a $\mathbb{R} \to \mathbb{R}$ $(k + 1)$-layer ReLU DNN of width $w$ such that if it is also representable by a $(k' + 1)$-layer ReLU DNN for any $k' \leq k$, then this $(k' + 1)$-layer ReLU DNN has size at least $\frac{1}{2}k'w^{\frac{k}{k'}} - 1$.

In fact our family of hard functions described above has a very intricate structure as stated below.

**Theorem 3.2.** For every $k \geq 1$, $w \geq 2$, every member of the family of hard functions in Theorem 3.1 has $w^k$ pieces and this family can be parametrized by

$$\bigcup_{M>0} \underbrace{(\Delta_M^{w-1} \times \Delta_M^{w-1} \times \ldots \times \Delta_M^{w-1})}_{k \text{ times}}, \tag{3.1}$$

i.e., for every point in the set above, there exists a distinct function with the stated properties.

The following is an immediate corollary of Theorem 3.1 by choosing the parameters carefully.

**Corollary 3.3.** For every $k \in \mathbb{N}$ and $\epsilon > 0$, there is a family of functions defined on the real line such that every function $f$ from this family can be represented by a $(k^{1+\epsilon}) + 1$-layer DNN with size $k^{2+\epsilon}$ and if $f$ is represented by a $k+1$-layer DNN, then this DNN must have size at least $\frac{1}{2}k \cdot k^{k^\epsilon} - 1$. Moreover, this family can be parametrized as, $\cup_{M>0} \Delta_M^{k^{2+\epsilon}-1}$.

A particularly illuminating special case is obtained by setting $\epsilon = 1$ in Corollary 3.3:

**Corollary 3.4.** For every natural number $k \in \mathbb{N}$, there is a family of functions parameterized by the set $\cup_{M>0} \Delta_M^{k^3-1}$ such that any $f$ from this family can be represented by a $k^2 + 1$-layer DNN with $k^3$ nodes, and every $k + 1$-layer DNN that represents $f$ needs at least $\frac{1}{2}k^{k+1} - 1$ nodes.

We can also get hardness of approximation versions of Theorem 3.1 and Corollaries 3.3 and 3.4, with the same gaps (upto constant terms), using the following theorem.

**Theorem 3.5.** For every $k \geq 1$, $w \geq 2$, there exists a function $f_{k,w}$ that can be represented by a $(k + 1)$-layer ReLU DNN with $w$ nodes in each layer, such that for all $\delta > 0$ and $k' \leq k$ the following holds:

$$\inf_{g \in \mathcal{G}_{k',\delta}} \int_{x=0}^1 |f_{k,w}(x) - g(x)|dx > \delta,$$

where $\mathcal{G}_{k',\delta}$ is the family of functions representable by ReLU DNNs with depth at most $k' + 1$, and size at most $k' \frac{w^{k/k'}(1-4\delta)^{1/k'}}{2^{1+1/k'}}$.

The depth-size trade-off results in Theorems 3.1, and 3.5 extend and improve Telgarsky's theorems from (Telgarsky, 2015; 2016) in the following three ways:

(i) If we use our Theorem 3.5 to the pair of neural nets considered by Telgarsky in Theorem 1.1 in Telgarsky (2016) which are at depths $k^3$ (of size also scaling as $k^3$) and $k$ then for this purpose of approximation in the $\ell_1-$norm we would get a size lower bound for the shallower net which scales as $\Omega(2^{k^2})$ which is exponentially (in depth) larger than the lower bound of $\Omega(2^k)$ that Telgarsky can get for this scenario.

(ii) Telgarsky's family of hard functions is parameterized by a single natural number $k$. In contrast, we show that for every *pair* of natural numbers $w$ and $k$, and a point from the set in equation 3.1, there exists a "hard" function which to be represented by a depth $k'$ network would need a size of at least $w^{\frac{k}{k'}} k'$. With the extra flexibility of choosing the parameter $w$, for the purpose of showing gaps in representation ability of deep nets we can shows size lower bounds which are *super*-exponential in depth as explained in Corollaries 3.3 and 3.4.

(iii) A characteristic feature of the "hard" functions in Boolean circuit complexity is that they are usually a countable family of functions and not a "smooth" family of hard functions. In fact, in the last section of Telgarsky (2015), Telgarsky states this as a "weakness" of the state-of-the-art results on "hard" functions for both Boolean circuit complexity and neural nets research. In contrast, we provide a smoothly parameterized family of "hard" functions in Section 3.1 (parametrized by the set in equation 3.1). Such a continuum of hard functions wasn't demonstrated before this work.

We point out that Telgarsky's results in (Telgarsky, 2016) apply to deep neural nets with a host of different activation functions, whereas, our results are specifically for neural nets with rectified linear units. In this sense, Telgarsky's results from (Telgarsky, 2016) are more general than our results in this paper, but with weaker gap guarantees. Eldan-Shamir (Shamir, 2016; Eldan & Shamir, 2016) show that there exists an $\mathbb{R}^n \to \mathbb{R}$ function that can be represented by a 3-layer DNN, that takes exponential in $n$ number of nodes to be approximated to within some constant by a 2-layer DNN. While their results are not immediately comparable with Telgarsky's or our results, it is an interesting open question to extend their results to a constant depth hierarchy statement analogous to the recent result of Rossman et al (Rossman et al., 2015). We also note that in last few years, there has been much effort in the community to show size lowerbounds on ReLU DNNs trying to approximate various classes of functions which are themselves not necessarily exactly representable by ReLU DNNs (Yarotsky, 2016; Liang & Srikant, 2016; Safran & Shamir, 2017).

## 3.2 A CONTINUUM OF HARD FUNCTIONS FOR $\mathbb{R}^n \to \mathbb{R}$ FOR $n \geq 2$

One measure of complexity of a family of $\mathbb{R}^n \to \mathbb{R}$ "hard" functions represented by ReLU DNNs is the asymptotics of the number of pieces as a function of dimension $n$, depth $k + 1$ and size $s$ of the ReLU DNNs. More precisely, suppose one has a family $\mathcal{H}$ of functions such that for every $n, k, w \in \mathbb{N}$ the family contains at least one $\mathbb{R}^n \to \mathbb{R}$ function representable by a ReLU DNN with depth at most $k + 1$ and maximum width at most $w$. The following definition formalizes a notion of complexity for such a $\mathcal{H}$.

**Definition 5** ($\mathrm{comp}_{\mathcal{H}}(n, k, w)$)**.** The measure $\mathrm{comp}_{\mathcal{H}}(n, k, w)$ is defined as the maximum number of pieces (see Definition 3) of a $\mathbb{R}^n \to \mathbb{R}$ function from $\mathcal{H}$ that can be represented by a ReLU DNN with depth at most $k + 1$ and maximum width at most $w$.

Similar measures have been studied in previous works Montufar et al. (2014); Pascanu et al. (2013); Raghu et al. (2016). The best known families $\mathcal{H}$ are the ones from Theorem 4 of (Montufar et al., 2014) and a mild generalization of Theorem 1.1 of (Telgarsky, 2016) to $k$ layers of ReLU activations with width $w$; these constructions achieve $\left( \lfloor (\frac{w}{n}) \rfloor \right)^{(k-1)n} (\sum_{j=0}^{n} \binom{w}{j})$ and $\mathrm{comp}_{\mathcal{H}}(n, k, s) = O(w^k)$, respectively. At the end of this section we would explain the precise sense in which we improve on these numbers. An analysis of this complexity measure is done using integer programming techniques in (Serra et al., 2017).

**Definition 6.** Let $\mathbf{b}^1, \ldots, \mathbf{b}^m \in \mathbb{R}^n$. The zonotope formed by $\mathbf{b}^1, \ldots, \mathbf{b}^m \in \mathbb{R}^n$ is defined as

$$Z(\mathbf{b}^1, \ldots, \mathbf{b}^m) := \{\lambda_1 \mathbf{b}^1 + \ldots + \lambda_m \mathbf{b}^m : -1 \leq \lambda_i \leq 1, \ i = 1, \ldots, m\}.$$

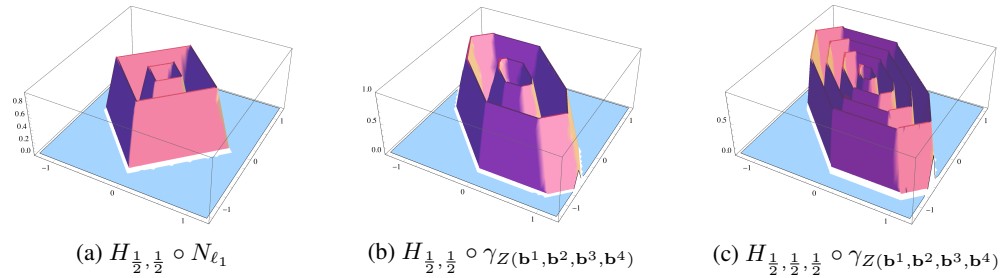

(a) $H_{\frac{1}{2},\frac{1}{2}} \circ N_{\ell_1}$    (b) $H_{\frac{1}{2},\frac{1}{2}} \circ \gamma_{Z(\mathbf{b}^1,\mathbf{b}^2,\mathbf{b}^3,\mathbf{b}^4)}$    (c) $H_{\frac{1}{2},\frac{1}{2},\frac{1}{2}} \circ \gamma_{Z(\mathbf{b}^1,\mathbf{b}^2,\mathbf{b}^3,\mathbf{b}^4)}$

Figure 1: We fix the $\mathbf{a}$ vectors for a two hidden layer $\mathbb{R} \to \mathbb{R}$ hard function as $\mathbf{a}^1 = \mathbf{a}^2 = (\frac{1}{2}) \in \Delta_1^1$ Left: A specific hard function induced by $\ell_1$ norm: $\text{ZONOTOPE}_{2,2,2}^2[\mathbf{a}^1,\mathbf{a}^2,\mathbf{b}^1,\mathbf{b}^2]$ where $\mathbf{b}^1 = (0,1)$ and $\mathbf{b}^2 = (1,0)$. Note that in this case the function can be seen as a composition of $H_{\mathbf{a}^1,\mathbf{a}^2}$ with $\ell_1$-norm $N_{\ell_1}(x) := \|x\|_1 = \gamma_{Z((0,1),(1,0))}$. Middle: A typical hard function $\text{ZONOTOPE}_{2,2,4}^2[\mathbf{a}^1,\mathbf{a}^2,\mathbf{c}^1,\mathbf{c}^2,\mathbf{c}^3,\mathbf{c}^4]$ with generators $\mathbf{c}^1 = (\frac{1}{4},\frac{1}{2}), \mathbf{c}^2 = (-\frac{1}{2},0), \mathbf{c}^3 = (0,-\frac{1}{4})$ and $\mathbf{c}^4 = (-\frac{1}{4},-\frac{1}{4})$. Note how increasing the number of zonotope generators makes the function more complex. Right: A *harder* function from $\text{ZONOTOPE}_{3,2,4}^2$ family with the same set of generators $\mathbf{c}_1, \mathbf{c}_2, \mathbf{c}_3, c_4$ but one more hidden layer ($k = 3$). Note how increasing the depth make the function more complex. (For illustrative purposes we plot only the part of the function which lies above zero.)

The set of vertices of $Z(\mathbf{b}^1,\dots,\mathbf{b}^m)$ will be denoted by $\text{vert}(Z(\mathbf{b}^1,\dots,\mathbf{b}^m))$. The *support function* $\gamma_{Z(\mathbf{b}^1,\dots,\mathbf{b}^m)} : \mathbb{R}^n \to \mathbb{R}$ associated with the zonotope $Z(\mathbf{b}^1,\dots,\mathbf{b}^m)$ is defined as

$$\gamma_{Z(\mathbf{b}^1,\dots,\mathbf{b}^m)}(\mathbf{r}) = \max_{\mathbf{x} \in Z(\mathbf{b}^1,\dots,\mathbf{b}^m)} \langle \mathbf{r}, \mathbf{x} \rangle.$$

The following results are well-known in the theory of zonotopes (Ziegler, 1995).

**Theorem 3.6.** The following are all true.

1. $|\text{vert}(Z(\mathbf{b}^1,\dots,\mathbf{b}^m))| \leq \sum_{i=0}^{n-1} \binom{m-1}{i}$. The set of $(\mathbf{b}^1,\dots,\mathbf{b}^m) \in \mathbb{R}^n \times \dots \times \mathbb{R}^n$ such that this *does not* hold at equality is a 0 measure set.

2. $\gamma_{Z(\mathbf{b}^1,\dots,\mathbf{b}^m)}(\mathbf{r}) = \max_{\mathbf{x} \in Z(\mathbf{b}^1,\dots,\mathbf{b}^m)} \langle \mathbf{r}, \mathbf{x} \rangle = \max_{\mathbf{x} \in \text{vert}(Z(\mathbf{b}^1,\dots,\mathbf{b}^m))} \langle \mathbf{r}, \mathbf{x} \rangle$, and $\gamma_{Z(\mathbf{b}^1,\dots,\mathbf{b}^m)}$ is therefore a piecewise linear function with $|\text{vert}(Z(\mathbf{b}^1,\dots,\mathbf{b}^m))|$ pieces.

3. $\gamma_{Z(\mathbf{b}^1,\dots,\mathbf{b}^m)}(\mathbf{r}) = |\langle \mathbf{r}, \mathbf{b}^1 \rangle| + \dots + |\langle \mathbf{r}, \mathbf{b}^m \rangle|$.

**Definition 7** (extremal zonotope set). The set $S(n,m)$ will denote the set of $(\mathbf{b}^1,\dots,\mathbf{b}^m) \in \mathbb{R}^n \times \dots \times \mathbb{R}^n$ such that $|\text{vert}(Z(\mathbf{b}^1,\dots,\mathbf{b}^m))| = \sum_{i=0}^{n-1} \binom{m-1}{i}$. $S(n,m)$ is the so-called "extremal zonotope set", which is a subset of $\mathbb{R}^{nm}$, whose complement has zero Lebesgue measure in $\mathbb{R}^{nm}$.

**Lemma 3.7.** Given any $\mathbf{b}^1,\dots,\mathbf{b}^m \in \mathbb{R}^n$, there exists a 2-layer ReLU DNN with size $2m$ which represents the function $\gamma_{Z(\mathbf{b}^1,\dots,\mathbf{b}^m)}(\mathbf{r})$.

**Definition 8.** For $p \in \mathbb{N}$ and $\mathbf{a} \in \Delta_M^p$, we define a function $h_{\mathbf{a}} : \mathbb{R} \to \mathbb{R}$ which is piecewise linear over the segments $(-\infty, 0], [0, \mathbf{a}_1], [\mathbf{a}_1, \mathbf{a}_2], \dots, [\mathbf{a}_p, M], [M, +\infty)$ defined as follows: $h_{\mathbf{a}}(x) = 0$ for all $x \leq 0$, $h_{\mathbf{a}}(\mathbf{a}_i) = M(i \mod 2)$, and $h_{\mathbf{a}}(M) = M - h_{\mathbf{a}}(\mathbf{a}_p)$ and for $x \geq M$, $h_{\mathbf{a}}(x)$ is a linear continuation of the piece over the interval $[\mathbf{a}_p, M]$. Note that the function has $p + 2$ pieces, with the leftmost piece having slope 0. Furthermore, for $\mathbf{a}^1,\dots,\mathbf{a}^k \in \Delta_M^p$, we denote the composition of the functions $h_{\mathbf{a}^1}, h_{\mathbf{a}^2}, \dots, h_{\mathbf{a}^k}$ by

$$H_{\mathbf{a}^1,\dots,\mathbf{a}^k} := h_{\mathbf{a}^k} \circ h_{\mathbf{a}^{k-1}} \circ \dots \circ h_{\mathbf{a}^1}.$$

**Proposition 3.8.** Given any tuple $(\mathbf{b}^1,\dots,\mathbf{b}^m) \in S(n,m)$ and any point

$$(\mathbf{a}^1,\dots,\mathbf{a}^k) \in \bigcup_{M>0} \underbrace{(\Delta_M^{w-1} \times \Delta_M^{w-1} \times \dots \times \Delta_M^{w-1})}_{k \text{ times}},$$

the function $\text{ZONOTOPE}_{k,w,m}^n[\mathbf{a}^1,\dots,\mathbf{a}^k,\mathbf{b}^1,\dots,\mathbf{b}^m] := H_{\mathbf{a}^1,\dots,\mathbf{a}^k} \circ \gamma_{Z(\mathbf{b}^1,\dots,\mathbf{b}^m)}$ has $(m-1)^{n-1}w^k$ pieces and it can be represented by a $k + 2$ layer ReLU DNN with size $2m + wk$.

Finally, we are ready to state the main result of this section.

**Theorem 3.9.** For every tuple of natural numbers $n, k, m \geq 1$ and $w \geq 2$, there exists a family of $\mathbb{R}^n \to \mathbb{R}$ functions, which we call $\text{ZONOTOPE}_{k,w,m}^n$ with the following properties:

(i) Every $f \in \text{ZONOTOPE}_{k,w,m}^n$ is representable by a ReLU DNN of depth $k + 2$ and size $2m + wk$, and has $\left( \sum_{i=0}^{n-1} \binom{m-1}{i} \right) w^k$ pieces.

(ii) Consider any $f \in \text{ZONOTOPE}_{k,w,m}^n$. If $f$ is represented by a $(k' + 1)$-layer DNN for any $k' \leq k$, then this $(k' + 1)$-layer DNN has size at least $\max \left\{ \frac{1}{2}(k' w^{\frac{k}{k'n}}) \cdot (m-1)^{(1-\frac{1}{n})\frac{1}{k'}} - 1 \ , \ \frac{w^{\frac{k}{k'}}}{n^{1/k'}} k' \right\}$.

(iii) The family $\text{ZONOTOPE}_{k,w,m}^n$ is in one-to-one correspondence with

$$S(n,m) \times \bigcup_{M>0} \underbrace{(\Delta_M^{w-1} \times \Delta_M^{w-1} \times \ldots \times \Delta_M^{w-1})}_{k \text{ times}}.$$

**Comparison to the results in (Montufar et al., 2014)**

*Firstly* we note that the construction in (Montufar et al., 2014) requires all the hidden layers to have width at least as big as the input dimensionality $n$. In contrast, we do not impose such restrictions and the network size in our construction is independent of the input dimensionality. Thus our result probes networks with bottleneck architectures whose complexity cant be seen from their result.

*Secondly*, in terms of our complexity measure, there seem to be regimes where our bound does better. One such regime, for example, is when $n \leq w < 2n$ and $k \in \Omega(\frac{n}{\log(n)})$, by setting in our construction $m < n$.

*Thirdly*, it is not clear to us whether the construction in (Montufar et al., 2014) gives a smoothly parameterized family of functions other than by introducing small perturbations of the construction in their paper. In contrast, we have a smoothly parameterized family which is in one-to-one correspondence with a well-understood manifold like the higher-dimensional torus.

## 4 Training 2-layer $\mathbb{R}^n \to \mathbb{R}$ ReLU DNNs to global optimality

In this section we consider the following empirical risk minimization problem. Given $D$ data points $(x_i, y_i) \in \mathbb{R}^n \times \mathbb{R}$, $i = 1, \ldots, D$, find the function $f$ represented by 2-layer $\mathbb{R}^n \to \mathbb{R}$ ReLU DNNs of width $w$, that minimizes the following optimization problem

$$\min_{f \in \mathcal{F}_{\{n,w,1\}}} \frac{1}{D} \sum_{i=1}^{D} \ell(f(x_i), y_i) \quad \equiv \quad \min_{T_1 \in \mathcal{A}_n^w, \, T_2 \in \mathcal{L}_w^1} \frac{1}{D} \sum_{i=1}^{D} \ell\big( T_2(\sigma(T_1(x_i))), y_i \big) \qquad (4.1)$$

where $\ell : \mathbb{R} \times \mathbb{R} \to \mathbb{R}$ is a convex *loss function* (common loss functions are the squared loss, $\ell(y, y') = (y - y')^2$, and the hinge loss function given by $\ell(y, y') = \max\{0, 1 - yy'\}$). Our main result of this section gives an algorithm to solve the above empirical risk minimization problem to global optimality.

**Theorem 4.1.** There exists an algorithm to find a global optimum of Problem 4.1 in time $O(2^w (D)^{nw} \text{poly}(D, n, w))$. Note that the running time $O(2^w (D)^{nw} \text{poly}(D, n, w))$ is polynomial in the data size $D$ for fixed $n, w$.

**Proof Sketch:** A full proof of Theorem 4.1 is included in Appendix C. Here we provide a sketch of the proof. When the empirical risk minimization problem is viewed as an optimization problem in the space of weights of the ReLU DNN, it is a nonconvex, quadratic problem. However, one can instead search over the space of functions representable by 2-layer DNNs by writing them in the form similar to (2.1). This breaks the problem into two parts: a combinatorial search and then a convex problem that is essentially linear regression with linear inequality constraints. This enables us to guarantee global optimality.

---

**Algorithm 1** Empirical Risk Minimization

---

```
1: function ERM(D)                                          ▷ Where D = {(x_i, y_i)}_{i=1}^D ⊂ R^n × R
2:     S = {+1, -1}^w                                       ▷ All possible instantiations of top layer weights
3:     P^i = {(P_+^i, P_-^i)},  i = 1, ..., w               ▷ All possible partitions of data into two parts
4:     P = P^1 × P^2 × ... × P^w
5:     count = 1                                                                              ▷ Counter
6:     for s ∈ S do
7:         for {(P_+^i, P_-^i)}_{i=1}^w ∈ P do
```

$$
8: \quad \text{loss(count)} = \begin{cases} \underset{\tilde{a}, \tilde{b}}{\text{minimize:}} & \sum_{j=1}^{D} \sum_{i:j \in P_+^i} \ell(s_i(\tilde{a}^i \cdot x_j + \tilde{b}_i), y_j) \\[2mm] \text{subject to:} & \tilde{a}^i \cdot x_j + \tilde{b}_i \leq 0 \quad \forall j \in P_-^i \\ & \tilde{a}^i \cdot x_j + \tilde{b}_i \geq 0 \quad \forall j \in P_+^i \end{cases}
$$

```
9:             count + +
10:        end for
11:        OPT = argmin loss(count)
12:    end for
13:    return {ã}, {b̃}, s corresponding to OPT's iterate
14: end function
```

---

Let $T_1(x) = Ax + b$ and $T_2(y) = a' \cdot y$ for $A \in \mathbb{R}^{w \times n}$ and $b, a' \in \mathbb{R}^w$. If we denote the $i$-th row of the matrix $A$ by $a^i$, and write $b_i, a_i'$ to denote the $i$-th coordinates of the vectors $b, a'$ respectively, due to homogeneity of ReLU gates, the network output can be represented as

$$
f(x) = \sum_{i=1}^{w} a_i' \max\{0, a^i \cdot x + b_i\} = \sum_{i=1}^{w} s_i \max\{0, \tilde{a}^i \cdot x + \tilde{b}_i\}.
$$

where $\tilde{a}^i \in \mathbb{R}^n$, $\tilde{b}_i \in \mathbb{R}$ and $s_i \in \{-1, +1\}$ for all $i = 1, \ldots, w$. For any hidden node $i \in \{1 \ldots, w\}$, the pair $(\tilde{a}^i, \tilde{b}_i)$ induces a partition $\mathcal{P}^i := (P_+^i, P_-^i)$ on the dataset, given by $P_-^i = \{j : \tilde{a}^i \cdot x_j + \tilde{b}_i \leq 0\}$ and $P_+^i = \{1, \ldots, D\} \backslash P_-^i$. Algorithm 1 proceeds by generating all combinations of the partitions $\mathcal{P}^i$ as well as the top layer weights $s \in \{+1, -1\}^w$, and minimizing the loss $\sum_{j=1}^{D} \sum_{i:j \in P_+^i} \ell(s_i(\tilde{a}^i \cdot x_j + \tilde{b}_i), y_j)$ subject to the constraints $\tilde{a}^i \cdot x_j + \tilde{b}_i \leq 0 \quad \forall j \in P_-^i$ and $\tilde{a}^i \cdot x_j + \tilde{b}_i \geq 0 \quad \forall j \in P_+^i$ which are imposed for all $i = 1, \ldots, w$, which is a convex program.

Algorithm 1 implements the empirical risk minimization (ERM) rule for training ReLU DNN with one hidden layer. To the best of our knowledge there is no other known algorithm that solves the ERM problem to global optimality. We note that due to known hardness results exponential dependence on the input dimension is unavoidable Blum & Rivest (1992); Shalev-Shwartz & Ben-David (2014); Algorithm 1 runs in time polynomial in the number of data points. To the best of our knowledge there is no hardness result known which rules out empirical risk minimization of deep nets in time polynomial in circuit size or data size. Thus our training result is a step towards resolving this gap in the complexity literature.

A related result for *improperly* learning ReLUs has been recently obtained by Goel et al (Goel et al., 2016). In contrast, our algorithm returns a ReLU DNN from the class being learned. Another difference is that their result considers the notion of *reliable learning* as opposed to the empirical risk minimization objective considered in (4.1).

## 5 DISCUSSION

The running time of the algorithm that we give in this work to find the exact global minima of a two layer ReLU-DNN is exponential in the input dimension $n$ and the number of hidden nodes $w$. The exponential dependence on $n$ can not be removed unless $P = NP$; see Shalev-Shwartz & Ben-David (2014); Blum & Rivest (1992); DasGupta et al. (1995). However, we are not aware of any complexity results which would rule out the possibility of an algorithm which trains to global optimality in time that is polynomial in the data size and/or the number of hidden nodes, assuming that the input dimension is a fixed constant. Resolving this dependence on network size would be another step towards clarifying the theoretical complexity of training ReLU DNNs and is a good

open question for future research, in our opinion. Perhaps an even better breakthrough would be to get optimal training algorithms for DNNs with two or more hidden layers and this seems like a substantially harder nut to crack. It would also be a significant breakthrough to get gap results between consecutive constant depths or between logarithmic and constant depths.

ACKNOWLEDGMENTS

We would like to thank Christian Tjandraatmadja for pointing out a subtle error in a previous version of the paper, which affected the complexity results for the number of linear regions in our constructions in Section 3.2. Anirbit would like to thank Ramprasad Saptharishi, Piyush Srivastava and Rohit Gurjar for extensive discussions on Boolean and arithmetic circuit complexity. This paper has been immensely influenced by the perspectives gained during those extremely helpful discussions. Amitabh Basu gratefully acknowledges support from the NSF grant CMMI1452820. Raman Arora was supported in part by NSF BIGDATA grant IIS-1546482.

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

## A  EXPRESSING PIECEWISE LINEAR FUNCTIONS USING ReLU DNNS

*Proof of Theorem 2.2.* Any continuous piecewise linear function $\mathbb{R} \to \mathbb{R}$ which has $m$ pieces can be specified by three pieces of information, (1) $s_L$ the slope of the left most piece, (2) the coordinates of the non-differentiable points specified by a $(m-1)-$tuple $\{(a_i, b_i)\}_{i=1}^{m-1}$ (indexed from left to right) and (3) $s_R$ the slope of the rightmost piece. A tuple $(s_L, s_R, (a_1, b_1), \dots, (a_{m-1}, b_{m-1}))$ uniquely specifies a $m$ piecewise linear function from $\mathbb{R} \to \mathbb{R}$ and vice versa. Given such a tuple, we construct a 2-layer DNN which computes the same piecewise linear function.

One notes that for any $a, r \in \mathbb{R}$, the function

$$f(x) = \begin{cases} 0 & x \le a \\ r(x-a) & x > a \end{cases} \tag{A.1}$$

is equal to $\operatorname{sgn}(r) \max\{|r|(x-a), 0\}$, which can be implemented by a 2-layer ReLU DNN with size 1. Similarly, any function of the form,

$$g(x) = \begin{cases} t(x-a) & x \le a \\ 0 & x > a \end{cases} \tag{A.2}$$

is equal to $-\operatorname{sgn}(t) \max\{-|t|(x-a), 0\}$, which can be implemented by a 2-layer ReLU DNN with size 1. The parameters $r, t$ will be called the *slopes* of the function, and $a$ will be called the *breakpoint* of the function.If we can write the given piecewise linear function as a sum of $m$ functions of the form (A.1) and (A.2), then by Lemma D.2 we would be done. It turns out that such a decomposition of any $p$ piece PWL function $h : \mathbb{R} \to \mathbb{R}$ as a sum of $p$ flaps can always be arranged where the breakpoints of the $p$ flaps all are all contained in the $p-1$ breakpoints of $h$. First, observe that adding a constant to a function does not change the complexity of the ReLU DNN expressing it, since this corresponds to a bias on the output node. Thus, we will assume that the value of $h$ at the last break point $a_{m-1}$ is $b_{m-1} = 0$. We now use a single function $f$ of the form (A.1) with slope $r$ and breakpoint $a = a_{m-1}$, and $m-1$ functions $g_1, \dots, g_{m-1}$ of the form (A.2) with slopes $t_1, \dots, t_{m-1}$ and breakpoints $a_1, \dots, a_{m-1}$, respectively. Thus, we wish to express $h = f + g_1 + \dots + g_{m-1}$. Such a decomposition of $h$ would be valid if we can find values for $r, t_1, \dots, t_{m-1}$ such that (1) the slope of the above sum is $= s_L$ for $x < a_1$, (2) the slope of the above sum is $= s_R$ for $x > a_{m-1}$, and (3) for each $i \in \{1, 2, 3, .., m-1\}$ we have $b_i = f(a_i) + g_1(a_i) + \dots + g_{m-1}(a_i)$.
The above corresponds to asking for the existence of a solution to the following set of simultaneous linear equations in $r, t_1, \dots, t_{m-1}$:

$$s_R = r, \quad s_L = t_1 + t_2 + \dots + t_{m-1}, \quad b_i = \sum_{j=i+1}^{m-1} t_j(a_{j-1} - a_j) \text{ for all } i = 1, \dots, m-2$$

It is easy to verify that the above set of simultaneous linear equations has a unique solution. Indeed, $r$ must equal $s_R$, and then one can solve for $t_1, \dots, t_{m-1}$ starting from the last equation $b_{m-2} =$

$t_{m-1}(a_{m-2} - a_{m-1})$ and then back substitute to compute $t_{m-2}, t_{m-3}, \ldots, t_1$. The lower bound of $p - 1$ on the size for any 2-layer ReLU DNN that expresses a $p$ piece function follows from Lemma D.6. $\qquad \square$

One can do better in terms of size when the rightmost piece of the given function is flat, i.e., $s_R = 0$. In this case $r = 0$, which means that $f = 0$; thus, the decomposition of $h$ above is of size $p - 1$. A similar construction can be done when $s_L = 0$. This gives the following statement which will be useful for constructing our forthcoming hard functions.

**Corollary A.1.** If the rightmost or leftmost piece of a $\mathbb{R} \to \mathbb{R}$ piecewise linear function has 0 slope, then we can compute such a $p$ piece function using a 2-layer DNN with size $p - 1$.

*Proof of theorem 2.3.* Since any piecewise linear function $\mathbb{R}^n \to \mathbb{R}$ is representable by a ReLU DNN by Corollary 2.1, the proof simply follows from the fact that the family of continuous piecewise linear functions is dense in any $L^p(\mathbb{R}^n)$ space, for $1 \leq p \leq \infty$. $\qquad \square$

# B   BENEFITS OF DEPTH

## B.1   CONSTRUCTING A CONTINUUM OF HARD FUNCTIONS FOR $\mathbb{R} \to \mathbb{R}$ ReLU DNNs AT EVERY DEPTH AND EVERY WIDTH

**Lemma B.1.** For any $M > 0$, $p \in \mathbb{N}$, $k \in \mathbb{N}$ and $\mathbf{a}^1, \ldots, \mathbf{a}^k \in \Delta_M^p$, if we compose the functions $h_{\mathbf{a}^1}, h_{\mathbf{a}^2}, \ldots, h_{\mathbf{a}^k}$ the resulting function is a piecewise linear function with at most $(p + 1)^k + 2$ pieces, i.e.,

$$H_{\mathbf{a}^1, \ldots, \mathbf{a}^k} := h_{\mathbf{a}^k} \circ h_{\mathbf{a}^{k-1}} \circ \ldots \circ h_{\mathbf{a}^1}$$

is piecewise linear with at most $(p+1)^k + 2$ pieces, with $(p+1)^k$ of these pieces in the range $[0, M]$ (see Figure 2). Moreover, in each piece in the range $[0, M]$, the function is affine with minimum value 0 and maximum value $M$.

*Proof.* Simple induction on $k$. $\qquad \square$

*Proof of Theorem 3.2.* Given $k \geq 1$ and $w \geq 2$, choose any point

$$(\mathbf{a}^1, \ldots, \mathbf{a}^k) \in \bigcup_{M > 0} \underbrace{(\Delta_M^{w-1} \times \Delta_M^{w-1} \times \ldots \times \Delta_M^{w-1})}_{k \text{ times}}.$$

By Definition 8, each $h_{\mathbf{a}^i}$, $i = 1, \ldots, k$ is a piecewise linear function with $w + 1$ pieces and the leftmost piece having slope 0. Thus, by Corollary A.1, each $h_{\mathbf{a}^i}$, $i = 1, \ldots, k$ can be represented by a 2-layer ReLU DNN with size $w$. Using Lemma D.1, $H_{\mathbf{a}^1, \ldots, \mathbf{a}^k}$ can be represented by a $k + 1$ layer DNN with size $wk$; in fact, each hidden layer has exactly $w$ nodes. $\qquad \square$

*Proof of Theorem 3.1.* Follows from Theorem 3.2 and Lemma D.6. $\qquad \square$

*Proof of Theorem 3.5.* Given $k \geq 1$ and $w \geq 2$ define $q := w^k$ and $s_q := \underbrace{h_{\mathbf{a}} \circ h_{\mathbf{a}} \circ \ldots \circ h_{\mathbf{a}}}_{k \text{ times}}$ where $\mathbf{a} = (\frac{1}{w}, \frac{2}{w}, \ldots, \frac{w-1}{w}) \in \Delta_1^{q-1}$. Thus, $s_q$ is representable by a ReLU DNN of width $w + 1$ and depth $k + 1$ by Lemma D.1. In what follows, we want to give a lower bound on the $\ell^1$ distance of $s_q$ from any continuous $p$-piecewise linear comparator $g_p : \mathbb{R} \to \mathbb{R}$. The function $s_q$ contains $\lfloor \frac{q}{2} \rfloor$ triangles of width $\frac{2}{q}$ and unit height. A $p$-piecewise linear function has $p - 1$ breakpoints in the interval $[0, 1]$. So that in at least $\lfloor \frac{w^k}{2} \rfloor - (p - 1)$ triangles, $g_p$ has to be affine. In the following we demonstrate that

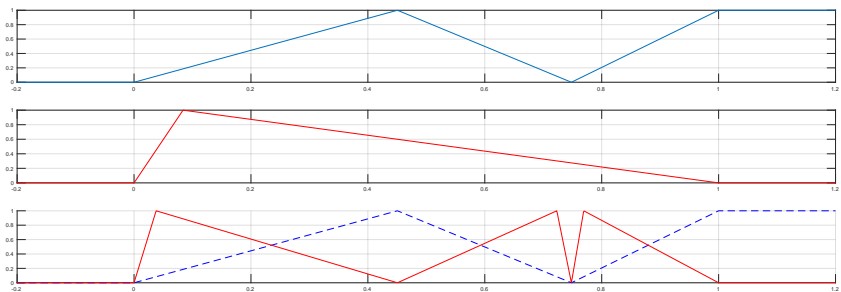

Figure 2: Top: $h_{\mathbf{a}^1}$ with $\mathbf{a}^1 \in \Delta_1^2$ with 3 pieces in the range $[0,1]$. Middle: $h_{\mathbf{a}^2}$ with $\mathbf{a}^2 \in \Delta_1^1$ with 2 pieces in the range $[0,1]$. Bottom: $H_{\mathbf{a}^1,\mathbf{a}^2} = h_{\mathbf{a}^2} \circ h_{\mathbf{a}^1}$ with $2 \cdot 3 = 6$ pieces in the range $[0,1]$. The dotted line in the bottom panel corresponds to the function in the top panel. It shows that for every piece of the dotted graph, there is a full copy of the graph in the middle panel.

inside any triangle of $s_q$, any affine function will incur an $\ell^1$ error of at least $\frac{1}{2w^k}$.

$$\int_{x=\frac{2i}{w^k}}^{\frac{2i+2}{w^k}} |s_q(x) - g_p(x)|dx = \int_{x=0}^{\frac{2}{w^k}} \left| s_q(x) - (y_1 + (x-0) \cdot \frac{y_2 - y_1}{\frac{2}{w^k} - 0}) \right| dx$$

$$= \int_{x=0}^{\frac{1}{w^k}} \left| xw^k - y_1 - \frac{w^k x}{2}(y_2 - y_1) \right| dx + \int_{x=\frac{1}{w^k}}^{\frac{2}{w^k}} \left| 2 - xw^k - y_1 - \frac{w^k x}{2}(y_2 - y_1) \right| dx$$

$$= \frac{1}{w^k} \int_{z=0}^{1} \left| z - y_1 - \frac{z}{2}(y_2 - y_1) \right| dz + \frac{1}{w^k} \int_{z=1}^{2} \left| 2 - z - y_1 - \frac{z}{2}(y_2 - y_1) \right| dz$$

$$= \frac{1}{w^k} \left( -3 + y_1 + \frac{2y_1^2}{2 + y_1 - y_2} + y_2 + \frac{2(-2+y_1)^2}{2 - y_1 + y_2} \right)$$

The above integral attains its minimum of $\frac{1}{2w^k}$ at $y_1 = y_2 = \frac{1}{2}$. Putting together,

$$\|s_{w^k} - g_p\|_1 \geq \left( \lfloor \frac{w^k}{2} \rfloor - (p-1) \right) \cdot \frac{1}{2w^k} \geq \frac{w^k - 1 - 2(p-1)}{4w^k} = \frac{1}{4} - \frac{2p-1}{4w^k}$$

Thus, for any $\delta > 0$,

$$p \leq \frac{w^k - 4w^k \delta + 1}{2} \implies 2p - 1 \leq (\frac{1}{4} - \delta)4w^k \implies \frac{1}{4} - \frac{2p-1}{4w^k} \geq \delta \implies \|s_{w^k} - g_p\|_1 \geq \delta.$$

The result now follows from Lemma D.6. $\qquad\square$

### B.2 A CONTINUUM OF HARD FUNCTIONS FOR $\mathbb{R}^n \to \mathbb{R}$ FOR $n \geq 2$

*Proof of Lemma 3.7.* By Theorem 3.6 part 3., $\gamma_{Z(\mathbf{b}^1,\ldots,\mathbf{b}^m)}(\mathbf{r}) = |\langle \mathbf{r}, \mathbf{b}^1 \rangle| + \ldots + |\langle \mathbf{r}, \mathbf{b}^m \rangle|$. It suffices to observe

$$|\langle \mathbf{r}, \mathbf{b}^1 \rangle| + \ldots + |\langle \mathbf{r}, \mathbf{b}^m \rangle| = \max\{\langle \mathbf{r}, \mathbf{b}^1 \rangle, -\langle \mathbf{r}, \mathbf{b}^1 \rangle\} + \ldots + \max\{\langle \mathbf{r}, \mathbf{b}^m \rangle, -\langle \mathbf{r}, \mathbf{b}^m \rangle\}.$$

$\qquad\square$

*Proof of Proposition 3.8.* The fact that $\text{ZONOTOPE}_{k,w,m}^n[\mathbf{a}^1, \ldots, \mathbf{a}^k, \mathbf{b}^1, \ldots, \mathbf{b}^m]$ can be represented by a $k+2$ layer ReLU DNN with size $2m + wk$ follows from Lemmas 3.7 and D.1. The number of pieces follows from the fact that $\gamma_{Z(\mathbf{b}^1,\ldots,\mathbf{b}^m)}$ has $\sum_{i=0}^{n-1} \binom{m-1}{i}$ distinct linear pieces by parts 1. and 2. of Theorem 3.6, and $H_{\mathbf{a}^1,\ldots,\mathbf{a}^k}$ has $w^k$ pieces by Lemma B.1. $\qquad\square$

*Proof of Theorem 3.9.* Follows from Proposition 3.8. $\qquad\square$

## C   EXACT EMPIRICAL RISK MINIMIZATION

*Proof of Theorem 4.1.* Let $\ell : \mathbb{R} \to \mathbb{R}$ be any convex loss function, and let $(x_1, y_1), \ldots, (x_D, y_D) \in \mathbb{R}^n \times \mathbb{R}$ be the given $D$ data points. As stated in (4.1), the problem requires us to find an affine transformation $T_1 : \mathbb{R}^n \to \mathbb{R}^w$ and a linear transformation $T_2 : \mathbb{R}^w \to \mathbb{R}$, so as to minimize the empirical loss as stated in (4.1). Note that $T_1$ is given by a matrix $A \in \mathbb{R}^{w \times n}$ and a vector $b \in \mathbb{R}^w$ so that $T(x) = Ax + b$ for all $x \in \mathbb{R}^n$. Similarly, $T_2$ can be represented by a vector $a' \in \mathbb{R}^w$ such that $T_2(y) = a' \cdot y$ for all $y \in \mathbb{R}^w$. If we denote the $i$-th row of the matrix $A$ by $a^i$, and write $b_i, a'_i$ to denote the $i$-th coordinates of the vectors $b, a'$ respectively, we can write the function represented by this network as

$$f(x) = \sum_{i=1}^{w} a'_i \max\{0, a^i \cdot x + b_i\} = \sum_{i=1}^{w} \operatorname{sgn}(a'_i) \max\{0, (|a'_i| a^i) \cdot x + |a'_i| b_i\}.$$

In other words, the family of functions over which we are searching is of the form

$$f(x) = \sum_{i=1}^{w} s_i \max\{0, \tilde{a}^i \cdot x + \tilde{b}_i\} \tag{C.1}$$

where $\tilde{a}^i \in \mathbb{R}^n$, $b_i \in \mathbb{R}$ and $s_i \in \{-1, +1\}$ for all $i = 1, \ldots, w$. We now make the following observation. For a given data point $(x_j, y_j)$ if $\tilde{a}^i \cdot x_j + \tilde{b}_i \leq 0$, then the $i$-th term of (C.1) does not contribute to the loss function for this data point $(x_j, y_j)$. Thus, for every data point $(x_j, y_j)$, there exists a set $S_j \subseteq \{1, \ldots, w\}$ such that $f(x_j) = \sum_{i \in S_j} s_i(\tilde{a}^i \cdot x_j + \tilde{b}_i)$. In particular, if we are given the set $S_j$ for $(x_j, y_j)$, then the expression on the right hand side of (C.1) reduces to a linear function of $\tilde{a}^i, \tilde{b}_i$. For any fixed $i \in \{1, \ldots, w\}$, these sets $S_j$ induce a partition of the data set into two parts. In particular, we define $P_+^i := \{j : i \in S_j\}$ and $P_-^i := \{1, \ldots, D\} \setminus P_+^i$. Observe now that this partition is also induced by the hyperplane given by $\tilde{a}^i, \tilde{b}_i$: $P_+^i = \{j : \tilde{a}^i \cdot x_j + \tilde{b}_i > 0\}$ and $P_+^i = \{j : \tilde{a}^i \cdot x_j + \tilde{b}_i \leq 0\}$. Our strategy will be to *guess* the partitions $P_+^i, P_-^i$ for each $i = 1, \ldots, w$, and then do linear regression with the constraint that regression's decision variables $\tilde{a}^i, \tilde{b}_i$ induce the guessed partition.

More formally, the algorithm does the following. For each $i = 1, \ldots, w$, the algorithm guesses a partition of the data set $(x_j, y_j), j = 1, \ldots, D$ by a hyperplane. Let us label the partitions as follows $(P_+^i, P_-^i), i = 1, \ldots, w$. So, for each $i = 1, \ldots, w$, $P_+^i \cup P_-^i = \{1, \ldots, D\}$, $P_+^i$ and $P_-^i$ are disjoint, and there exists a vector $c \in \mathbb{R}^n$ and a real number $\delta$ such that $P_-^i = \{j : c \cdot x_j + \delta \leq 0\}$ and $P_+^i = \{j : c \cdot x_j + \delta > 0\}$. Further, for each $i = 1, \ldots, w$ the algorithm selects a vector $s$ in $\{+1, -1\}^w$.

For a fixed selection of partitions $(P_+^i, P_-^i), i = 1, \ldots, w$ and a vector $s$ in $\{+1, -1\}^w$, the algorithm solves the following convex optimization problem with decision variables $\tilde{a}^i \in \mathbb{R}^n$, $\tilde{b}_i \in \mathbb{R}$ for $i = 1, \ldots, w$ (thus, we have a total of $(n + 1) \cdot w$ decision variables). The feasible region of the optimization is given by the constraints

$$\begin{aligned} \tilde{a}^i \cdot x_j + \tilde{b}_i \leq 0 \quad &\forall j \in P_-^i \\ \tilde{a}^i \cdot x_j + \tilde{b}_i \geq 0 \quad &\forall j \in P_+^i \end{aligned} \tag{C.2}$$

which are imposed for all $i = 1, \ldots, w$. Thus, we have a total of $D \cdot w$ constraints. Subject to these constraints we minimize the objective $\sum_{j=1}^{D} \sum_{i:j \in P_+^i} \ell(s_i(\tilde{a}^i \cdot x_j + \tilde{b}_i), y_j)$. Assuming the loss function $\ell$ is a convex function in the first argument, the above objective is a convex function. Thus, we have to minize a convex objective subject to the linear inequality constraints from (C.2).

We finally have to count how many possible partitions $(P_+^i, P_-^i)$ and vectors $s$ the algorithm has to search through. It is well-known Matousek (2002) that the total number of possible hyperplane partitions of a set of size $D$ in $\mathbb{R}^n$ is at most $2\binom{D}{n} \leq D^n$ whenever $n \geq 2$. Thus with a guess for each $i = 1, \ldots, w$, we have a total of at most $D^{nw}$ partitions. There are $2^w$ vectors $s$ in $\{-1, +1\}^w$. This gives us a total of $2^w D^{nw}$ guesses for the partitions $(P_+^i, P_-^i)$ and vectors $s$. For each such guess, we have a convex optimization problem with $(n + 1) \cdot w$ decision variables and $D \cdot w$ constraints, which can be solved in time $\operatorname{poly}(D, n, w)$. Putting everything together, we have the running time claimed in the statement.

The above argument holds only for $n \geq 2$, since we used the inequality $2\binom{D}{n} \leq D^n$ which only holds for $n \geq 2$. For $n = 1$, a similar algorithm can be designed, but one which uses the characterization achieved in Theorem 2.2. Let $\ell : \mathbb{R} \to \mathbb{R}$ be any convex loss function, and let $(x_1, y_1), \ldots, (x_D, y_D) \in \mathbb{R}^2$ be the given $D$ data points. Using Theorem 2.2, to solve problem (4.1) it suffices to find a $\mathbb{R} \to \mathbb{R}$ piecewise linear function $f$ with $w$ pieces that minimizes the total loss. In other words, the optimization problem (4.1) is equivalent to the problem

$$\min\left\{\sum_{i=1}^{D} \ell(f(x_i), y_i) : f \text{ is piecewise linear with } w \text{ pieces}\right\}. \tag{C.3}$$

We now use the observation that fitting piecewise linear functions to minimize loss is just a step away from linear regression, which is a special case where the function is contrained to have exactly one affine linear piece. Our algorithm will first guess the optimal partition of the data points such that all points in the same class of the partition correspond to the same affine piece of $f$, and then do linear regression in each class of the partition. Altenatively, one can think of this as guessing the interval $(x_i, x_{i+1})$ of data points where the $w - 1$ breakpoints of the piecewise linear function will lie, and then doing linear regression between the breakpoints.

More formally, we parametrize piecewise linear functions with $w$ pieces by the $w$ slope-intercept values $(a_1, b_1), \ldots, (a_2, b_2), \ldots, (a_w, b_w)$ of the $w$ different pieces. This means that between breakpoints $j$ and $j+1$, $1 \leq j \leq w - 2$, the function is given by $f(x) = a_{j+1}x + b_{j+1}$, and the first and last pieces are $a_1x + b_1$ and $a_wx + b_w$, respectively.

Define $\mathcal{I}$ to be the set of all $(w-1)$-tuples $(i_1, \ldots, i_{w-1})$ of natural numbers such that $1 \leq i_1 \leq \ldots \leq i_{w-1} \leq D$. Given a fixed tuple $I = (i_1, \ldots, i_{w-1}) \in \mathcal{I}$, we wish to search through all piecewise linear functions whose breakpoints, in order, appear in the intervals $(x_{i_1}, x_{i_1+1}), (x_{i_2}, x_{i_2+1})$, $\ldots, (x_{i_{w-1}}, x_{i_{w-1}+1})$. Define also $\mathcal{S} = \{-1, 1\}^{w-1}$. Any $S \in \mathcal{S}$ will have the following interpretation: if $S_j = 1$ then $a_j \leq a_{j+1}$, and if $S_j = -1$ then $a_j \geq a_{j+1}$. Now for every $I \in \mathcal{I}$ and $S \in \mathcal{S}$, requiring a piecewise linear function that respects the conditions imposed by $I$ and $S$ is easily seen to be equivalent to imposing the following linear inequalities on the parameters $(a_1, b_1), \ldots, (a_2, b_2), \ldots, (a_w, b_w)$:

$$\begin{aligned} S_j(b_{j+1} - b_j - (a_j - a_{j+1})x_{i_j}) &\geq 0 \\ S_j(b_{j+1} - b_j - (a_j - a_{j+1})x_{i_j+1}) &\leq 0 \\ S_j(a_{j+1} - a_j) &\geq 0 \end{aligned} \tag{C.4}$$

Let the set of piecewise linear functions whose breakpoints satisfy the above be denoted by $\mathrm{PWL}^1_{I,S}$ for $I \in \mathcal{I}, S \in \mathcal{S}$.

Given a particular $I \in \mathcal{I}$, we define

$$\begin{aligned} D_1 &:= \{x_i : i \leq i_1\}, \\ D_j &:= \{x_i : i_{j-1} < i \leq i_1\} \quad j = 2, \ldots, w-1, \\ D_w &:= \{x_i : i > i_{w-1}\} \end{aligned}$$

Observe that

$$\min\{\sum_{i=1}^{D} \ell(f(x_i)-y_i) : f \in \mathrm{PWL}^1_{I,S}\} = \min\{\sum_{j=1}^{w}\left(\sum_{i \in D_j} \ell(a_j \cdot x_i + b_j - y_i)\right) : (a_j, b_j) \text{ satisfy (C.4)}\} \tag{C.5}$$

The right hand side of the above equation is the problem of minimizing a convex objective subject to linear constraints. Now, to solve (C.3), we need to simply solve the problem (C.5) for all $I \in \mathcal{I}, S \in \mathcal{S}$ and pick the minimum. Since $|\mathcal{I}| = \binom{D}{w} = O(D^w)$ and $|\mathcal{S}| = 2^{w-1}$ we need to solve $O(2^w \cdot D^w)$ convex optimization problems, each taking time $O(\mathrm{poly}(D))$. Therefore, the total running time is $O((2D)^w \mathrm{poly}(D))$.

$\square$

## D  Auxiliary Lemmas

Now we will collect some straightforward observations that will be used often. The following operations preserve the property of being representable by a ReLU DNN.

**Lemma D.1.** [Function Composition] If $f_1 : \mathbb{R}^d \to \mathbb{R}^m$ is represented by a $d, m$ ReLU DNN with depth $k_1 + 1$ and size $s_1$, and $f_2 : \mathbb{R}^m \to \mathbb{R}^n$ is represented by an $m, n$ ReLU DNN with depth $k_2 + 1$ and size $s_2$, then $f_2 \circ f_1$ can be represented by a $d, n$ ReLU DNN with depth $k_1 + k_2 + 1$ and size $s_1 + s_2$.

*Proof.* Follows from (1.1) and the fact that a composition of affine transformations is another affine transformation. □

**Lemma D.2.** [Function Addition] If $f_1 : \mathbb{R}^n \to \mathbb{R}^m$ is represented by a $n, m$ ReLU DNN with depth $k + 1$ and size $s_1$, and $f_2 : \mathbb{R}^n \to \mathbb{R}^m$ is represented by a $n, m$ ReLU DNN with depth $k + 1$ and size $s_2$, then $f_1 + f_2$ can be represented by a $n, m$ ReLU DNN with depth $k + 1$ and size $s_1 + s_2$.

*Proof.* We simply put the two ReLU DNNs in parallel and combine the appropriate coordinates of the outputs. □

**Lemma D.3.** [Taking maximums/minimums] Let $f_1, \ldots, f_m : \mathbb{R}^n \to \mathbb{R}$ be functions that can each be represented by $\mathbb{R}^n \to \mathbb{R}$ ReLU DNNs with depths $k_i + 1$ and size $s_i$, $i = 1, \ldots, m$. Then the function $f : \mathbb{R}^n \to \mathbb{R}$ defined as $f(\mathbf{x}) := \max\{f_1(\mathbf{x}), \ldots, f_m(\mathbf{x})\}$ can be represented by a ReLU DNN of depth at most $\max\{k_1, \ldots, k_m\} + \log(m) + 1$ and size at most $s_1 + \ldots s_m + 4(2m - 1)$. Similarly, the function $g(\mathbf{x}) := \min\{f_1(\mathbf{x}), \ldots, f_m(\mathbf{x})\}$ can be represented by a ReLU DNN of depth at most $\max\{k_1, \ldots, k_m\} + \lceil \log(m) \rceil + 1$ and size at most $s_1 + \ldots s_m + 4(2m - 1)$.

*Proof.* We prove this by induction on $m$. The base case $m = 1$ is trivial. For $m \geq 2$, consider $g_1 := \max\{f_1, \ldots, f_{\lfloor \frac{m}{2} \rfloor}\}$ and $g_2 := \max\{f_{\lfloor \frac{m}{2} \rfloor + 1}, \ldots, f_m\}$. By the induction hypothesis (since $\lfloor \frac{m}{2} \rfloor, \lceil \frac{m}{2} \rceil < m$ when $m \geq 2$), $g_1$ and $g_2$ can be represented by ReLU DNNs of depths at most $\max\{k_1, \ldots, k_{\lfloor \frac{m}{2} \rfloor}\} + \lceil \log(\lfloor \frac{m}{2} \rfloor) \rceil + 1$ and $\max\{k_{\lfloor \frac{m}{2} \rfloor + 1}, \ldots, k_m\} + \lceil \log(\lceil \frac{m}{2} \rceil) \rceil + 1$ respectively, and sizes at most $s_1 + \ldots s_{\lfloor \frac{m}{2} \rfloor} + 4(2\lfloor \frac{m}{2} \rfloor - 1)$ and $s_{\lfloor \frac{m}{2} \rfloor + 1} + \ldots + s_m + 4(2\lfloor \frac{m}{2} \rfloor - 1)$, respectively. Therefore, the function $G : \mathbb{R}^n \to \mathbb{R}^2$ given by $G(\mathbf{x}) = (g_1(\mathbf{x}), g_2(\mathbf{x}))$ can be implemented by a ReLU DNN with depth at most $\max\{k_1, \ldots, k_m\} + \lceil \log(\lceil \frac{m}{2} \rceil) \rceil + 1$ and size at most $s_1 + \ldots + s_m + 4(2m - 2)$.

We now show how to represent the function $T : \mathbb{R}^2 \to \mathbb{R}$ defined as $T(x, y) = \max\{x, y\} = \frac{x+y}{2} + \frac{|x-y|}{2}$ by a 2-layer ReLU DNN with size 4 – see Figure 3. The result now follows from the fact that $f = T \circ G$ and Lemma D.1. □

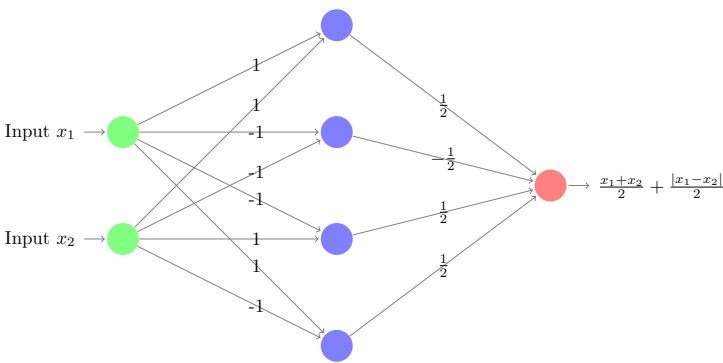

Figure 3: A 2-layer ReLU DNN computing $\max\{x_1, x_2\} = \frac{x_1+x_2}{2} + \frac{|x_1-x_2|}{2}$

**Lemma D.4.** Any affine transformation $T : \mathbb{R}^n \to \mathbb{R}^m$ is representable by a 2-layer ReLU DNN of size $2m$.

*Proof.* Simply use the fact that $T = (I \circ \sigma \circ T) + (-I \circ \sigma \circ (-T))$, and the right hand side can be represented by a 2-layer ReLU DNN of size $2m$ using Lemma D.2. □

**Lemma D.5.** Let $f : \mathbb{R} \to \mathbb{R}$ be a function represented by a $\mathbb{R} \to \mathbb{R}$ ReLU DNN with depth $k + 1$ and widths $w_1, \ldots, w_k$ of the $k$ hidden layers. Then $f$ is a PWL function with at most $2^{k-1} \cdot (w_1 + 1) \cdot w_2 \cdot \ldots \cdot w_k$ pieces.

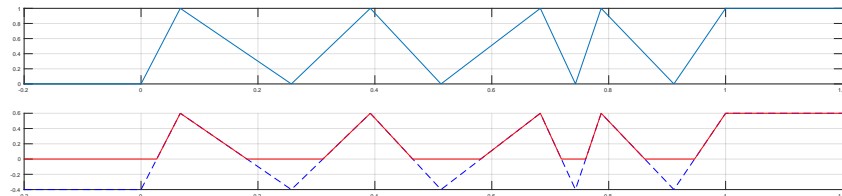

Figure 4: The number of pieces increasing after activation. If the blue function is $f$, then the red function $g = \max\{0, f + b\}$ has at most twice the number of pieces as $f$ for any bias $b \in \mathbb{R}$.

*Proof.* We prove this by induction on $k$. The base case is $k = 1$, i.e, we have a 2-layer ReLU DNN. Since every activation node can produce at most one breakpoint in the piecewise linear function, we can get at most $w_1$ breakpoints, i.e., $w_1 + 1$ pieces.

Now for the induction step, assume that for some $k \geq 1$, any $\mathbb{R} \to \mathbb{R}$ ReLU DNN with depth $k + 1$ and widths $w_1, \ldots, w_k$ of the $k$ hidden layers produces at most $2^{k-1} \cdot (w_1 + 1) \cdot w_2 \cdot \ldots \cdot w_k$ pieces.

Consider any $\mathbb{R} \to \mathbb{R}$ ReLU DNN with depth $k + 2$ and widths $w_1, \ldots, w_{k+1}$ of the $k + 1$ hidden layers. Observe that the input to any node in the last layer is the output of a $\mathbb{R} \to \mathbb{R}$ ReLU DNN with depth $k + 1$ and widths $w_1, \ldots, w_k$. By the induction hypothesis, the input to this node in the last layer is a piecewise linear function $f$ with at most $2^{k-1} \cdot (w_1 + 1) \cdot w_2 \cdot \ldots \cdot w_k$ pieces. When we apply the activation, the new function $g(x) = \max\{0, f(x)\}$, which is the output of this node, may have at most twice the number of pieces as $f$, because each original piece may be intersected by the $x$-axis; see Figure 4. Thus, after going through the layer, we take an affine combination of $w_{k+1}$ functions, each with at most $2 \cdot (2^{k-1} \cdot (w_1 + 1) \cdot w_2 \cdot \ldots \cdot w_k)$ pieces. In all, we can therefore get at most $2 \cdot (2^{k-1} \cdot (w_1 + 1) \cdot w_2 \cdot \ldots \cdot w_k) \cdot w_{k+1}$ pieces, which is equal to $2^k \cdot (w_1 + 1) \cdot w_2 \cdot \ldots \cdot w_k \cdot w_{k+1}$, and the induction step is completed. □

Lemma D.5 has the following consequence about the depth and size tradeoffs for expressing functions with agiven number of pieces.

**Lemma D.6.** Let $f : \mathbb{R} \to \mathbb{R}$ be a piecewise linear function with $p$ pieces. If $f$ is represented by a ReLU DNN with depth $k + 1$, then it must have size at least $\frac{1}{2} k p^{1/k} - 1$. Conversely, any piecewise linear function $f$ that represented by a ReLU DNN of depth $k + 1$ and size at most $s$, can have at most $(\frac{2s}{k})^k$ pieces.

*Proof.* Let widths of the $k$ hidden layers be $w_1, \ldots, w_k$. By Lemma D.5, we must have

$$2^{k-1} \cdot (w_1 + 1) \cdot w_2 \cdot \ldots \cdot w_k \geq p. \tag{D.1}$$

By the AM-GM inequality, minimizing the size $w_1 + w_2 + \ldots + w_k$ subject to (D.1), means setting $w_1 + 1 = w_2 = \ldots = w_k$. This implies that $w_1 + 1 = w_2 = \ldots = w_k \geq \frac{1}{2} p^{1/k}$. The first statement follows. The second statement follows using the AM-GM inequality again, this time with a restriction on $w_1 + w_2 + \ldots + w_k$. □

