# OpenReview forum: "Understanding Deep Neural Networks with Rectified Linear Units"
_ICLR.cc/2018/Conference — Accept (Poster)_

### Official Review · AnonReviewer3 · 2017-11-26
**Review of "Understanding Deep Neural Networks with Rectified Linear Units"**

**Rating:** 6
**Confidence:** 4

**Review:**

This paper presents several theoretical results regarding the expressiveness and learnability of ReLU-activated deep neural networks. I summarize the main results as below:

(1) Any piece-wise linear function can be represented by a ReLU-acteivated DNN. Any smooth function can be approximated by such networks.

(2) The expressiveness of 3-layer DNN is stronger than any 2-layer DNN.

(3) Using a polynomial number of neurons, the ReLU-acteivated DNN can represent a piece-wise linear function with exponentially many pieces

(4) The ReLU-activated DNN can be learnt to global optimum with an exponential-time algorithm.

Among these results (1), (2), (4) are sort of known in the literature. This paper extends the existing results in some subtle ways. For (1), the authors show that the DNN has a tighter bound on the depth. For (2), the "hard" functions has a better parameterization, and the gap between 3-layer and 2-layer is proved bigger. For (4), although the algorithm is exponential-time, it guarantees to compute the global optimum.

The stronger results of (1), (2), (4) all rely on the specific piece-wise linear nature of ReLU. Other than that, I don't get much more insight from the theoretical result. When the input dimension is n, the representability result of (1) fails to show that a polynomial number of neurons is sufficient. Perhaps an exponential number of neurons is necessary in the worst case, but it will be more interesting if the authors show that under certain conditions a polynomial-size network is good enough.

Result (3) is more interesting as it is a new result. The authors present a constructive proof to show that ReLU-activated DNN can represent many linear pieces.  However, the construction seems artificial and these functions don't seem to be visually very complex.

Overall, this is an incremental work in the direction of studying the representation power of neural networks. The results might be of theoretical interest, but I doubt if a pragmatic ReLU network user will learn anything by reading this paper.

---

### Official Review · AnonReviewer1 · 2017-11-27
**This paper investigates the function classes representable by ReLU networks. It contributes a more detailed view on hard functions representable by deep networks, their parametrisation, and gaps between deep and shallow.**

**Rating:** 6
**Confidence:** 5

**Review:**

The paper presents a series of definitions and results elucidating details about the functions representable by ReLU networks, their parametrisation, and gaps between deep and shallower nets.

The paper is easy to read, although it does not seem to have a main focus (exponential gaps vs. optimisation vs. universal approximation). The paper makes a nice contribution to the details of deep neural networks with ReLUs, although I find the contributed results slightly overstated. The 1d results are not difficult to derive from previous results. The advertised new results on the asymptotic behaviour assume a first layer that dominates the size of the network. The optimisation method appears close to brute force and is limited to 2 layers.

Theorem 3.1 appears to be easily deduced from the results from Montufar, Pascanu, Cho, Bengio, 2014. For 1d inputs, each layer will multiply the number of regions at most by the number of units in the layer, leading to the condition w’ \geq w^{k/k’}. Theorem 3.2 is simply giving a parametrization of the functions, removing symmetries of the units in the layers.

In the list at the top of page 5. Note that, the function classes might be characterized in terms of countable properties, such as the number of linear regions as discussed in MPCB, but still they build a continuum of functions. Similarly, in page 5 ``Moreover, for fixed n,k,s, our functions are smoothly parameterized''. This should not be a surprise.

In the last paragraph of Section 3 ``m = w^k-1'' This is a very big first layer. This also seems to subsume the first condition, s\geq  w^k-1 +w(k-1) for the network discussed in Theorem 3.9. In the last paragraph of Section 3 ``To the best of our knowledge''. In the construction presented here, the network’s size is essentially in the layer of size m. Under such conditions, Corollary 6 of MPCB also reads as s^n. Here it is irrelevant whether one artificially increases the depth of the network by additional, very narrow, layers, which do not contribute to the asymptotic number of units.

The function class Zonotope is a composition of two parts. It would be interesting to consider also a single construction, instead of the composition of two constructions.

Theorem 3.9 (ii) it would be nice to have a construction where the size becomes 2m + wk when k’=k.

Section 4, while interesting, appears to be somewhat disconnected from the rest of the paper.

In Theorem 2.3. explain why the two layer case is limited to n=1.

At some point in the first 4 pages it would be good to explain what is meant by ``hard’’ functions (e.g. functions that are hard to represent, as opposed to step functions, etc.)

---

### Official Review · AnonReviewer2 · 2017-11-29
**good paper, consider publishing**

**Rating:** 7
**Confidence:** 4

**Review:**

The paper presents an analysis and characterization of ReLU networks (with a linear final layer) via the set of functions these networks can model, especially focusing on the set of “hard” functions that are not easily representable by shallower networks.  It makes several important contributions, including extending the previously published bounds by Telgarsky et al. to tighter bounds for the special case of ReLU DNNs, giving a construction for a family of hard functions whose affine pieces scale exponentially with the dimensionality of the inputs, and giving a procedure for searching for globally optimal solution of a 1-hidden layer ReLU DNN with linear output layer and convex loss.  I think these contributions warrant publishing the paper at ICLR 2018.  The paper is also well written, a bit dense in places, but overall well organized and easy to follow.

A key limitation of the paper in my opinion is that typically DNNs do not contain a linear final layer.  It will be valuable to note what, if any, of the representation analysis and global convergence results carry over to networks with non-linear (Softmax, e.g.) final layer.  I also think that the global convergence algorithm is practically unfeasible for all but trivial use cases due to terms like D^nw, would like hearing authors’ comments in case I’m missing some simplification.

One minor suggestion for improving readability is to explicitly state, whenever applicable, that functions under consideration are PWL.  For example, adding PWL to Theorems and Corollaries in Section 3.1 will help.  Similarly would be good to state, wherever applicable, the DNN being discussed is a ReLU DNN.

---

### Decision · Program_Chairs · 2018-01-29
**ICLR 2018 Conference Acceptance Decision**

**Decision:**

Accept (Poster)

**Comment:**

Theoretical analysis and understanding of DNNs is a crucial area for ML community. This paper studies characteristics of the relu DNNs and makes several important contributions.